# Exploring Training Time Modality Incompleteness and Learning from Diverse Modalities

## Abstract

Multimodal learning benefits from the complementary signals across different data sources, but real-world scenarios often encounter missing modalities, particularly during training. Existing approaches focus on addressing this issue at test time and typically rely on fully co-occurring multimodal data, which can be difficult and costly to collect. We propose a two-stage framework designed to address training-time modality incompleteness without requiring co-occurring samples. The first stage, Data Fusing with Label-guided Mapping (DFLM), constructs a pseudo-multimodal dataset by aligning user data across modalities using supervised contrastive learning guided by shared labels. The second stage, Cooperative Cross-attention Multimodal Transformer (CCAMT), learns from the constructed dataset using a cross-attention mechanism that supports both modality-specific learning and cross-modal interaction with drastically different modalities. An extensive evaluation on three popular datasets (Multimodal Twitter, Multimodal Reddit, and StudentLife) demonstrates that CCAMT significantly outperforms the best-published baselines across all metrics. CCAMT achieves an impressive 96.5% accuracy, significantly outperforming single-modal baselines by up to 10.5% in accuracy. The physical activity data increases the model's accuracy by 2.8%. It also significantly outperforms the state-of-the-art time2vec multimodal transformer by 3% in accuracy, 2.9% in F1 score, 0.9% in precision, and 2.8% in recall. It outperforms other strong multimodal baselines by up to a 7.7% increase in accuracy and a 6.8% improvement in F1 score. Our robustness analysis with imbalanced data evaluation shows that CCAMT can achieve 74.2% accuracy with only 10% of data, significantly outperforming Time2Vec Transformer (at 47.3%) and SetTransformer (at 50.2%). The Edge deployment evaluation also shows that CCAMT's encoder configuration is up to 83.04% faster than other configurations on an Nvidia Jetson device.

## 1 Introduction

Multimodal learning has achieved notable success across a wide range of tasks by leveraging complementary information from diverse sources. Applications such as multimodal sentiment analysis (Rahman et al., 2020), depression detection (Gui et al., 2019b), and visual question answering (Antol et al., 2015) demonstrate that combining modalities like text, audio, and images can substantially boost performance by capturing different aspects of the target phenomenon.

Despite this promise, real-world deployments often face partial modality availability. For example, in mental health monitoring or user behavior modeling, observable inputs such as social media posts or profile images may not be sufficient to capture internal states like mood or stress level. These hidden factors often require additional sensing modalities, such as physiological or behavioral data, for reliable inference. However, acquiring meaningful modalities datasets with all co-occurring modalities is often costly, intrusive, and difficult to scale. These limitations highlight a practical yet underexplored question: *How can we enable effective multimodal learning without co-occurring modality data?*

To solve this problem, we need to address the following two challenges. First, how to create a multimodal dataset from single-modal datasets containing drastically different modalities without requiring co-occurring samples? Researchers have proposed solutions to tackle this modality unavailability problem, but they primarily focus on test-time settings. Modality hallucination (Hoffman et al., 2016) trains an auxiliary network to mimic depth features from RGB input. Modality distillation (Garcia et al., 2018) transfers information from depth to RGB via feature- and label-based supervision. MissModal (Lin & Hu, 2023) aligns modal-complete and modal-incomplete inputs using contrastive and distribution losses. These approaches are insufficient when some modalities are unavailable during training time, as is often the case in real-world settings.

The second challenge is how to effectively learn from a multimodal dataset with drastically different modalities? Existing multimodal learning techniques focus on areas where there is a clearly defined, well-studied shared semantic space, such as visual question answering with image and text (Antol et al., 2015) and multimodal sentiment analysis with video, audio, and text (Tsai et al., 2019). However, many real-world applications involve diverse combinations of modalities, including time-series data (e.g., physiological or behavioral signals), non-time-series data (e.g., aggregated statistics or categorical inputs), which are less studied in the multimodal learning literature.

To address these challenges, we propose a two-stage framework that consists of: 1) Data Fusing with Label-guided Mapping (DFLM), which constructs a pseudo-multimodal dataset by augmenting missing modalities using existing user data based on shared labels, and 2) Cooperative Cross-Attention Multimodal Transformer (CCAMT), which learns from the resulting dataset by a cooperative cross-attention mechanism that enables both modality-specific learning and cross-modal interaction with drastically different modalities. *To the best of our knowledge, this is the first framework to address training-time modality incompleteness through a data fusion approach and the first to learn from a dataset with drastically different modalities.* As a case study, we consider the early depression detection task using online activity data (including image and text) and physical activity data (including 12 modalities, such as step counts and phone usage).

To construct the pseudo-multimodal dataset, DFLM fuses physical activity data and online activity data based on semantic alignment in the embedding space. DFLM maps online and physical modality embeddings into a shared latent space using lightweight projection networks trained via contrastive learning to align semantically similar users. It first converts raw data into modality-specific embeddings with transformer-based encoders. For online activity data, it uses pre-trained models such as CLIP (Radford et al., 2021) and EmoBerta (Kim & Vossen, 2021) to extract image and text embeddings. For physical activity data, it leverages our proposed modality-tailored feature extraction technique that transforms diverse but distinct data characteristic-preserving and uniform representation, which is then encoded by a unified transformer. DFLM generates pseudo-multimodal user pairs through similarity-based matching in the shared latent space. Each pair forms a pseudo sample containing both online and physical modality embeddings. Aggregating these samples produces a pseudo-multimodal dataset with complementary, semantically aligned signals. This dataset enables downstream multimodal learning even in the absence of co-occurring data.

To learn from this pseudo-multimodal dataset, we propose the Cooperative Cross-Attention Multimodal Transformer (CCAMT). It effectively fuses complementary information across drastically different but semantically aligned modalities. At its core is our cooperative cross-attention encoder, which enables each modality to attend to others for cross-modal interaction while also attending to different parts of itself to improve its representation. Following cross-attention encoding, a final transformer encoder aggregates the fused representations and outputs the prediction. This architecture enables CCAMT to effectively leverage pseudo-multimodal signals for accurate downstream classification.

We comprehensively evaluate our approach on three popular datasets: Multimodal Twitter, Multimodal Reddit, and StudentLife, and find that CCAMT consistently outperforms the best-published related works across all evaluated metrics. We apply the proposed data fusion technique to augment the missing physical activity data type to the online activity datasets and then learn from them. On Multimodal Twitter, CCAMT achieves 96.5% accuracy, 96.3% F1 score, 96.2% precision, and 96.4% recall, outperforming the state-of-the-art baseline by 2.78% in accuracy, 2.79% in F1 score, 1.06% in precision, and 2.23% in recall. On Multimodal Reddit, it reaches 93.5% accuracy, surpassing the state-of-the-art Time2vec Transformer by 2.9%. Our robustness analysis further confirms CCAMT's robustness under extreme data imbalance. It consistently outperforms the baselines,

achieving 74.2% with only 10% data, significantly outperforming Time2vec transformer (49.8%) and SetTransformer (50.2%). For edge deployment, we evaluate six encoder configurations and confirms the feasibility of providing timely and accurate predictions on Nvidia Jetson devices. Our chosen encoder combination, Clip + EmoBERTa, achieves the lowest inference latency of 10.75 seconds, up to 83.04% faster than other configurations.

In this paper, we explore a practical alternative to co-occurring multimodal data by creating pseudo-multimodal datasets and designing a model to learn from them effectively. Our main contributions are as follows: 1) a contrastive data-fusing method that creates pseudo-multimodal datasets by aligning disjoint user data across modalities based on embedding level similarity and label consistency; 2) a cooperative cross-attention transformer that enables effective learning from the generated pseudo-multimodal datasets; and 3) comprehensive evaluations across diverse datasets and deployment settings, demonstrating the effectiveness and robustness of our framework.

## 2 BACKGROUND AND RELATED WORKS

**Multimodal depression detection.** Multimodal approaches leverage diverse data sources to improve depression detection. An et al. (An et al., 2020) combined text and speech features, and Gui et al. (Gui et al., 2019b) added image features to further improve performance. Dominguez et al. (Domínguez-Jiménez et al., 2020) integrated physiological data such as heart rate and electrodermal activity. Sun et al. (Sun et al., 2020) fused visual data with facial expressions and physiological data, resulting in a better understanding of mental states. Despite these advances, most existing work focuses on temporally aligned modalities and overlooks the potential of combining physical and online activity data. In practice, these two data types offer complementary perspectives on user behavior but are rarely studied together. To address this, our proposed framework constructs pseudo-multimodal datasets considering data with drastically different modalities and employs a cooperative cross-attention multimodal transformer to learn from this dataset.

**Cross-modal consistency in depression.** Mental states such as depression manifest in both physical and online behaviors, including reduced mobility, irregular sleep, and less engaging communication. Prior studies demonstrate this cross-modal consistency. Sobin et al. (Sobin & Sackeim, 2025) reported reduced physical activity in depressed individuals, and Chancellor et al. (Chancellor & De Choudhury, 2020) observed decreased online activities. These patterns suggest that individuals with similar mental states often exhibit coherent behavioral signals across modalities. Due to the lack of datasets that contain co-occurring physical and online activity data, our data-fusing technique bridges this gap by reusing real user single-modal data to construct pseudo-multimodal datasets.

**Modality Incompleteness.** Prior work has explored missing modalities at both test time and training time. At test time, models typically hallucinate or distill missing information. Hoffman et al. (Hoffman et al., 2016) introduced modality hallucination to mimic depth features from RGB. Garcia et al. (Garcia et al., 2018) proposed modality distillation (Garcia et al., 2018), where a hallucination network transfers depth knowledge to an RGB model. MissModal (Lin & Hu, 2023) improves robustness by aligning complete and incomplete modality representations through contrastive and distribution-based objectives. Our approach shares the same underlying principle as prior works, but it addresses a different problem of training time modality incompleteness.

Training-time incompleteness presents a different challenge. Fortin et al. (Fortin & Chaib-Draa, 2019) proposed a multi-task framework with unimodal and multimodal classifiers that can still be trained when some modalities are absent. Ma et al. (Ma et al., 2021) introduced a Bayesian meta-learning approach that enables single-modality embeddings to approximate full-modality ones, even under severe missingness. These approaches still rely on at least some paired multimodal data, whereas our method operates without *any* co-occurrence and instead constructs a pseudo-multimodal dataset by fusing heterogeneous single-modal sources.

**Supervised contrastive learning.** Traditional supervised contrastive learning forms positive pairs by grouping samples that share the same class label, encouraging models to pull their embeddings together in the latent space (Khosla et al., 2020). This approach assumes all the data comes from the same modality. In comparison, we apply supervised contrastive learning across different modalities, using shared mental health labels to align user embeddings from online and physical activity data.

**Comparison to multimodal diffusion.** CMMD (Yang et al., 2024) and our method both leverage contrastive learning but address fundamentally different multimodal problems. CMMD operates on paired video–audio data, but even with temporal alignment, the two encoders can produce semantically misaligned features because the modalities capture different cues. CMMD addresses this by using a contrastive diffusion loss that pulls the paired latent representations into a shared semantic space. In contrast, our framework addresses a completely different challenge: multimodal learning when modalities do not co-occur and come from disjoint user groups. Our method uses a contrastive objective to to align semantically similar user data across heterogeneous single-modal datasets and construct a pseudo multimodal dataset.

## 3 METHODOLOGY

Our proposed a two-stage framework consists of: 1) Data Fusing with Label-guided Mapping (DFLM), which constructs a pseudo-multimodal dataset by combining existing datasets with drastically different modalities, and 2) Cooperative Cross-Attention Multimodal Transformer (CCAMT), which learns from the resulting dataset. We will discuss the details in the following sections.

### 3.1 DATA FUSING WITH LABEL GUIDED MAPPING

Data Fusing with Label Guided Mapping (DFLM) leverages supervised contrastive learning to align semantically similar user data across heterogeneous single-modal datasets and construct a pseudo-multimodal dataset. It first extracts modality-specific embeddings and trains lightweight projection networks to map them into a shared latent space. It then applies a similarity-based matching strategy to align users across modalities. As a case study, we use online and physical activity data, which differ in both content and structure, to explain these steps.

**Modality-specific embedding extraction.** We consider three modalities, denoted as $\alpha$, $\beta$, and $\gamma$, each providing an input feature sequence: $X^\alpha \in \mathbb{R}^{T_\alpha \times d_\alpha}$, $X^\beta \in \mathbb{R}^{T_\beta \times d_\beta}$, and $X^\gamma \in \mathbb{R}^{T_\gamma \times d_\gamma}$, where $T_m$ is the sequence length and $d_m$ is the feature dimension for modality $m \in \{\alpha, \beta, \gamma\}$. To extract modality-specific embeddings, we use pre-trained transformer encoders $f_\alpha$, $f_\beta$, and $f_\gamma$, resulting in: $E^\alpha = f_\alpha X^\alpha \in \mathbb{R}^{T \times d'}$, $E^\beta = f_\beta X^\beta \in \mathbb{R}^{T \times d'}$, $E^\gamma = f_\gamma X^\gamma \in \mathbb{R}^{T \times d'}$, where $d'$ denotes the output embedding dimension of the encoders. These extracted embeddings serve two purposes. First, our framework directly uses them in downstream multimodal learning: $\hat{y} = F(E^\alpha, E^\beta, E^\gamma)$, where $F$ is a multimodal transformer and $\hat{y}$ is the model output (e.g., prediction label). Second, our framework projects these embeddings into a shared latent space to facilitate our proposed data-fusing.

The physical activity data $\gamma$ includes diverse modalities (e.g., GPS, accelerometer, and sleep patterns), each capturing distinct aspects of user behavior. To simplify integration, we treat these as a single modality through a two-stage process: we first extract modality-specific statistical and behavioral features, then concatenate them into a unified sequence. This sequence is encoded using a transformer to produce the embedding $E^\gamma$. This design preserves the unique characteristics of each modality and enables a transformer encoder to generate embeddings for downstream learning.

**Training pairs construction.** To train the projection networks with contrastive learning, DFLM constructs training pairs using pre-extracted embeddings from online and physical user data. For each online user $u$, we randomly assign a physical user $v$ sampled from the pool of users with the same depressive label. For the online modality, we use a projection network $g_{\text{online}}$ on the concatenated embeddings: $z^{\text{online}} = g_{\text{online}} E^{\text{online}} \in \mathbb{R}^{T \times d''}$. For the physical modality, we apply a separate network: $z^{\text{physical}} = g_{\text{physical}} E^{\text{physical}} \in \mathbb{R}^{T \times d''}$. Both projection networks are lightweight MLPs with identical architectures. Our preprocessing step ensures that the online modality's encoder outputs a fixed sequence length. The physical modality does not naturally match this length, so we apply a 1D interpolation step to resize its sequence to match it.

**Optimizing contrastive objective.** We train the projection networks $g_{\text{online}}$ and $g_{\text{physical}}$ using the standard InfoNCE loss, which encourages paired online and physical embeddings to be close in the shared space while pushing apart unpaired samples. Each projector consists of two linear layers with a ReLU activation in between: $z = \text{MLP}(x) = W_2 \text{ReLU}(W_1 x + b_1) + b_2$, where $x$ is the input embedding and $z$ is the projected output. Given a batch of $B$ online embeddings $\{z_i^{\text{online}}\}_{i=1}^B$

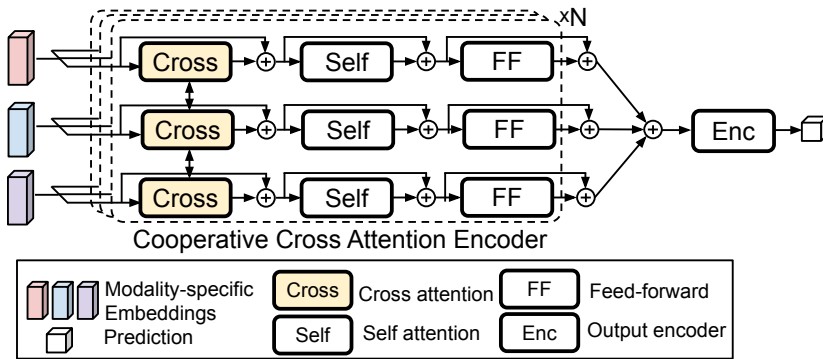

Figure 1: Overall architecture of Cooperative Cross-Attention Multimodal Transformer.

and physical embeddings $\{z_i^{\text{physical}}\}_{i=1}^{B}$, we define the contrastive loss as follows:

$$\mathcal{L}_{\text{InfoNCE}} = \frac{1}{B} \sum_{i=1}^{B} - \log \frac{\exp(\text{sim}(z_i^{\text{online}}, z_i^{\text{physical}})/\tau)}{\sum_{j=1}^{B} \exp(\text{sim}(z_i^{\text{online}}, z_j^{\text{physical}})/\tau)}, \tag{1}$$

where $\text{sim}(\cdot, \cdot)$ denotes the cosine similarity between two embeddings, and $\tau$ is a temperature hyperparameter controlling the smoothness of the similarity scores. $z_i^{\text{online}}$ and $z_i^{\text{physical}}$ form a positive pair (i.e., online and physical share the same depressive label) and all $z_j^{\text{physical}}$ for $j \neq i$ serve as negative examples drawn from the batch.

**Similarity-based matching.** DFLM then matches online and physical users based on their projected embedding similarities to create the pseudo-multimodal dataset. It first projects all online and physical user embeddings into the shared latent space using the trained projection networks $g_{\text{online}}$ and $g_{\text{physical}}$. To ensure semantic consistency, DFLM divides the projected physical embeddings into groups based on their original depressive labels. For each online user, we consider only physical users with the same label as potential matches. Using these curated label groups, DFLM computes cosine similarities between the online embedding $z_i^{\text{online}}$ and all the candidate physical embeddings $\{z_j^{\text{physical}}\}$ in the same group. It assigns the best-matching physical user by selecting the one with the highest similarity: $j^* = \arg\max_j \frac{z_i^{\text{online}} \cdot z_j^{\text{physical}}}{\|z_i^{\text{online}}\| \|z_j^{\text{physical}}\|}$, where $j^*$ is the index of the selected physical user. This matching procedure results in a pseudo-multimodal dataset, where each example combines online data with an augmented physical modality. Although the original modalities do not co-occur, the resulting dataset is label-aligned and semantically meaningful. In the next section, we introduce a transformer-based model specifically designed to learn from such a dataset.

### 3.2 COOPERATIVE CROSS ATTENTION MULTIMODAL TRANSFORMER

**Model overview.** Our proposed Cooperative Cross Attention Multimodal Transformer (CCAMT) integrates text, image, and physical modalities for downstream prediction. At its core is a cooperative cross-attention encoder that enables cross-modal information exchange and representation learning, followed by a final transformer encoder and a classification head.

**Attention layer details.** We implement the cooperative cross-attention encoder using two types of layers: cross-attention, which enables information exchange across modalities, and self-attention, which captures intra-modality dependencies. As a foundation, we build on the attention mechanism (Vaswani et al., 2017), a core element of transformer architectures that allows the model to selectively attend to relevant parts of the input, capturing long-range and contextual dependencies. The following description explains the attention mechanism (Vaswani et al., 2017): $\text{Attention}(Q, K, V) = \text{softmax}\left(\frac{QK^T}{\sqrt{d_k}}\right)V$, where $Q$ (query), $K$ (key), and $V$ (value) are projections of the input embeddings, which are dense vectors. $d_k$ is the key vector dimension; its square root normalizes the dot product of $Q$ and $K$ to stabilize gradients softmax output.

This attention mechanism serves as the foundation for the proposed cooperative cross-attention encoder. We build it using a stack of $N$ identical layers, each containing cross-attention, self-attention, and feed-forward sub-layers. Cross-attention enables information exchange between modalities by focusing on modality pairs. Self-attention allows the model to attend to different parts of the same modality. The feed-forward applies non-linear transformations with two fully connected layers and an activation function in between. Each sub-layer employs a residual connection and layer normalization at the output, as indicated by the '+' sign in Figure 1.

To illustrate the cooperative cross-attention mechanism, we introduce the following notations. Let $m_i^{x,k-1}$ and $m_j^{y,k-1}$ represent the embeddings from modality $x$ and $y$ at layer $(k-1)$. Attention scores are then computed between them to model inter-modal influence. For cross-attention between modality $x$ and $y$, queries $Q$ are the projections of modality $x's$ embeddings, and keys $K$ and values $V$ are the projections of modality $y's$ embedding. $W_Q, W_K$, and $W_V$ are learned weights:

$$Q_{xy} = W_Q\{m_i^{x,k-1}\}, K_{xy} = W_K\{m_j^{y,k-1}\}, V_{xy} = W_V\{m_j^{y,k-1}\}. \tag{2}$$

This allows the model to align and integrate information from modality $x$ to modality $y$. Similarly, for cross-attention from modality $y$ to $x$, the model uses embeddings of modality $y$ as queries and embeddings of modality $x$ as keys and values:

$$Q_{yx} = W_Q\{m_j^{y,k-1}\}, K_{yx} = W_K\{m_i^{x,k-1}\}, V_{yx} = W_V\{m_i^{x,k-1}\}, \tag{3}$$

After obtaining each set of $Q, K$, and $V$, the model computes the attention scores as follows for each modality pair: Attention Scores$_{xy} = \text{softmax}\left(\frac{Q_{xy}K_{xy}^T}{\sqrt{d_k}}\right)$. The model then applies these attention scores to the corresponding value ($V$) matrix to obtain the final output of the attention mechanism for each modality pair: Output$_{xy} = $ Attention Scores$_{xy} \cdot V_{xy}$.

**Output transformer and final prediction.** The encoder aggregates all pairwise attention outputs into a unified representation $H^{\text{aggregated}} \in \mathbb{R}^{T \times d}$, which is then passed into an output transformer encoder. We apply mean pooling to the transformer's output, followed by a fully connected layer and sigmoid activation to produce the final prediction: $\hat{y} = \sigma\left(W_c \cdot \text{MeanPool}(\text{Transformer}(H^{\text{aggregated}}))\right)$, where $\sigma(\cdot)$ denotes the sigmoid activation for binary classification and $W_c$ denotes the the final fully-connected layer's weights.

## 4 EVALUATION

**Scope and dataset suitability.** Most existing multimodal sentiment datasets (e.g., MOSI (Zadeh et al., 2016), MOSEI (Zadeh et al., 2018), CMU-MOSEAS (Zadeh et al., 2020)) provide co-occurring modalities such as aligned video, audio, and text. This setup differs from the non-co-occurring, cross-user scenario we study, where modalities come from separate sources and share only label-level supervision. As a result, these datasets do not naturally support evaluating our cooperative alignment setting.

**Datasets.** An ideal depression dataset for benchmarking CCAMT should capture both physical and online activity. However, public datasets contain only one activity type, limiting models from capturing full-spectrum user behavior. To address this, we use our proposed data-fusing technique to augment public online and physical activity datasets. We evaluate performance using standard metrics: accuracy, precision, recall, and F1 score.

**Online activity datasets.** We evaluate on two widely used multimodal depression datasets. *Multimodal Twitter* (Gui et al., 2019b) extends the textual depression dataset by Shen et al. (Shen et al., 2017), consisting of 691K tweets with images and text from 1,402 depressed and 1,402 control users. We follow Gui et al. (Gui et al., 2019b) and use an 80:20 train–test split. *Multimodal Reddit* (Bucur et al., 2023) contains 1,419 users from the depressed and 2,344 control users. We follow the original train/val/test split of 2,633, 379, and 751 users for fair comparison.

**Physical activity dataset.** *StudentLife*. This dataset (Wang et al., 2014) consists of physical activity data collected from 30 undergraduate and 18 graduate students at Dartmouth over a 10-week term. It includes continuous smartphone sensing data. such as sleep patterns and physical activity. It also

contains 32,000 daily self-reports on mood, stress, and loneliness, along with pre-post surveys such as PHQ-9 (Kroenke et al., 2001) and the UCLA loneliness scale (Russell, 1996).

**Baseline models.** We categorize our baselines into three groups based on their characteristics: transformer-based models, conventional models, and hybrid models.

*Transformer-based models* use deep-learning architectures with self-attention mechanisms. Our proposed CCAMT model belongs to this category, along with baselines like EmoBerta Transformers (Kim & Vossen, 2021), Vanilla Transformers (Bucur et al., 2023), and Time2Vec Transformer (Bucur et al., 2023). EmoBERTa Transformer (Kim & Vossen, 2021) incorporates linguistic and emotional cues; Vanilla Transformers (Bucur et al., 2023) is a multimodal transformer that uses the learned positional embedding method, and Time2Vec Transformer (T2V) (Bucur et al., 2023) uses time-enriched positional embedding (Kazemi et al., 2019). MulT (Tsai et al., 2019) is a multimodal transformer that aligns and fuses information across modalities through cross-modal attention.

To include MulT as a three-modality baseline, we implemented it within our two-stage pipeline. The original MulT targets synchronized audio–video–text streams and cannot be directly integrated into our framework because: 1) our modalities come from heterogeneous encoders and therefore lack the consistent low-level temporal structure assumed in MulT, and 2) our pipeline operates on sequences are resampled into a unified temporal window with a fixed masking format that MulT does not natively support. To address this, we preserved its core components, including temporal convolution alignment and directional cross-modal attention. We then convert each modality's embeddings into MulT's expected input structure and applied our unified fixed-window masks to make its alignment layers operate correctly within our pipeline.

*Conventional models* use traditional deep-learning architectures such as long short-term memory (LSTM) and convolutional neural networks (CNN) for depression detection. Time-aware LSTM (Baytas et al., 2017) (T-LSTM) integrates time information in the memory unit of LSTM, while LSTM + RL (Gui et al., 2019a) and CNN + RL (Gui et al., 2019a) apply reinforcement learning to identify posts that reflect users' depression behavior.

*Hybrid models* combine two commonly used model architectures. This category includes Multimodal Topic-enriched Auxiliary Learning (MTAL) (An et al., 2020), which uses the multimodal topic information for depression detection, and GRU + VGG-Net + COMMA (Gui et al., 2019b), which integrates GRU, VGG-Net, and reinforcement learning to select depression-indicative posts from text and images.

**Image and text encoders.** We evaluate various pre-trained encoders for CCAMT. For images, we use CLIP (Radford et al., 2021) for its strong vision-language alignment, and DINO (Caron et al., 2021) for its performance on downstream tasks. For text, we consider three transformers: RoBERTa (Liu et al., 2019), which excels in downstream tasks; EmoBerta (Kim & Vossen, 2021), which captures both linguistic and emotional nuances; and Multilingual MiniLM (Wang et al., 2020), a distilled version of XLM-RoBERTa (Conneau et al., 2019). See Appendix A.1 for implementation details.

Table 1: Performance comparison between multimodal and single-modal models.

| Dataset | Method | Modality | F1 | Recall | Precision | Accuracy |
|---------|--------|----------|------|------|------|------|
| Twitter | T-LSTM | T | 85.6 | 91.6 | 81.4 | 86 |
| | EmoBERTa Transformer | T | 87.4 | 91.7 | 85.3 | 87.1 |
| | LSTM + RL | T | 87.1 | 87 | 87.2 | 87 |
| | CNN + RL | T | 87.1 | 87.1 | 87.1 | 87.1 |
| | Time2VecTransformer | T+I | 93.5 | 94.1 | 95.1 | 93.5 |
| | CCAMT w/o physical | T+I | 93.3 | 94.1 | 92.7 | 93.3 |
| | MulT | T+I+P | 95.6 | 95.6 | 95.5 | 95.6 |
| | CCAMT (Ours) | T+I+P | **96.4** | 96 | **96.9** | **96.5** |
| Reddit | T-LSTM (Alt) | T | 85.1 | 83.3 | 85.7 | 87.9 |
| | EmoBERTa Transformer | T | 84.9 | 86.8 | 83.1 | 88.3 |
| | Time2VecTransformer | T+I | 87.6 | 87.7 | 87.6 | 90.6 |
| | MulT | T+I+P | 92.2 | **94.6** | 89.8 | 92.0 |
| | CCAMT (Ours) | T+I+P | **93.6** | 93 | **94.2** | **93.5** |

### 4.1 COMPARING TO SINGLE-MODAL AND LATE-FUSION METHODS

First, to confirm the effectiveness of our multimodal approach, we compare the proposed CCAMT against single-modal models and late-fusion baselines.

**Comparison multimodal and single-modal results.** Table 1 demonstrates that CCAMT surpasses single-modal baselines, such as T-LSTM (Baytas et al., 2017) and CNN (Gui et al., 2019a), which

are trained only on textual online data. CCAMT significantly improves accuracy by up to 10.5% and F1 score by up to 10.8%. This confirms our model's ability to comprehend complex modal interdependencies among different modalities, which single-modal models fail to capture. We also evaluate the impact of physical activity data on the model's performance. Table 1 demonstrates that when the physical activity data is removed ("CCAMT w/o physical data"), the F1 score drops from 96.4 to 93.3, comparable to the baseline Time2VecTransformer's F1 score of 93.5. This confirms the effectiveness of incorporating physical activity data on the model's performance. CCAMT also outperforms MulT with 0.8 and 1.4 F1 improvements, respectively, demonstrating stronger cross-modal performance.

**Comparison to late-fusion methods.** To further evaluate the effectiveness of CCAMT in multimodal learning, we compare it against a late-fusion baseline that aggregates the outputs of three independently trained single-modal transformers (see Table 2). We adopt the single-modal transformer design from prior work (Bucur et al., 2023), training one model per modality and applying late fusion

Table 2: Performance comparison between late-fusion methods.

| Method | Modality | F1 | Recall | Precision | Accuracy |
|---|---|---|---|---|---|
| Single-modal Transformer | T | 78.4 | 92.1 | 68.3 | 74.6 |
| Single-modal Transformer | I | 68.0 | 95.0 | 53.0 | 55.2 |
| Single-modal Transformer | P | 72.5 | 92.5 | 59.7 | 65.0 |
| Single-modal Transformer LF | T+I+P | 74.6 | 64.1 | 89.2 | 74.2 |
| CCAMT (Ours) | T+I+P | **96.4** | **96.0** | **96.9** | **96.5** |

at the output level (Single-modal Transformer LF). CCAMT outperforms this late-fusion baseline significantly, improving accuracy by 22.3%.

## 4.2 COMPARISON TO SOTA MULTIMODAL METHODS

Next, to evaluate the effectiveness of the proposed Cooperative Cross-Attention Multimodal Transformer (CCAMT) in learning from complex, multimodal data, we compare it against the best-published results in depression detection using the Multimodal Twitter dataset. Table 3 compares the results, and CCAMT exhibits superior performance across all evaluated metrics when compared against a wide range of baselines.

**Comparison to transformer-based models.**

For the multimodal Twitter dataset, CCAMT achieves an accuracy of 96.5%, significantly outperforming the state-of-the-art time2vec multimodal transformer by 3% in accuracy, 2.9% in F1 score, 0.9% in precision, and 2.8% in recall. When compared to other multimodal transformers, such as the Vanilla Transformer and SetTransformer, CCAMT shows up to a 7.7% increase in accuracy, a 6.8% improvement in F1 score, a 7.2% improvement in precision, and a 4.4% boost in recall. Furthermore, when compared to the single-modal EmoBERTa Transformer,

Table 3: Overall performance comparison on the Multimodal Twitter and Multimodal Reddit.

| Dataset | Method | F1 | Recall | Precision | Accuracy |
|---|---|---|---|---|---|
| Multimodal Twitter | MTAL | 84.2 | 84.2 | 84.2 | 84.2 |
| | GRU + VGG + COMMA | 90.0 | 90.1 | 90.0 | 90.0 |
| | MTAN | 90.8 | 93.1 | 88.5 | - |
| | Vanilla Transformer | 89.6 | 92.5 | 88.8 | 88.8 |
| | SetTransformer | 93.5 | 95.4 | 93.1 | 92.9 |
| | Time2VecTransformer | 93.5 | 94.1 | 95.1 | 93.5 |
| | CCAMT (Ours) | **96.4** | **96.0** | **96.9** | **96.5** |
| Multimodal Reddit | Uban et al. | - | - | - | 66.3 |
| | VanillaTransformer | 84.5 | 85.8 | 83.7 | 88.2 |
| | SetTransformer | 90.9 | 93.8 | 88.4 | 92.9 |
| | Time2VecTransformer | 87.6 | 87.7 | 87.6 | 90.6 |
| | CCAMT (Ours) | **93.6** | **93.0** | **94.2** | **93.5** |

CCAMT surpasses it by a large margin: 9.4% higher accuracy, 9.0% higher F1 score, 10.7% better precision, and a 5.2% improvement in recall. These results confirm the efficacy of our model in learning from multimodal data.

For the multimodal Reddit dataset, CCAMT also outperforms baselines across all metrics. It achieves 93.6% F1 score, 94.2% recall, 92.9% precision, and 93.5% accuracy. It outperforms the Time2vec Transformer by 3% and the SetTransformer by 0.7%. Compared to the single-modal EmoBERTa Transformer, CCAMT demonstrates a significant 8.7% accuracy improvement. Additionally, it outperforms traditional T-LSTM by 8.5% in accuracy. These results confirm the efficacy of our model in learning from multimodal data.

**Comparison to hybrid multimodal models.** CCAMT achieves markedly better performance compared to hybrid models such as MTAL (An et al., 2020) and the combination of GRU, VGG-Net, and COMMA (GRU + VGG + COMMA) (Gui et al., 2019b). Specifically, CCAMT achieves an improvement in F1 scores by as much as 12.2%.

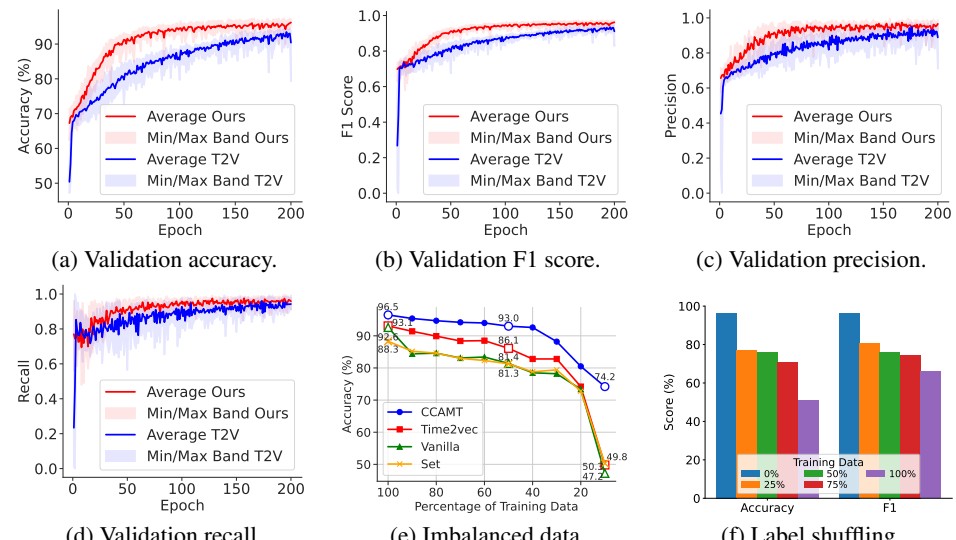

Figure 2: The progression of the model's performance and test accuracy comparison. Solid lines represent the averaged results across five folds, with fluctuations visualized through the Min/Max Band, showing the minimum and maximum values for each fold. (d) Test accuracy of CCAMT with baselines on varying percentages of depressed users and (e) Label shuffling results.

Overall, transformer-based models, especially CCAMT, demonstrate the highest accuracy, significantly outperforming hybrid multimodal and conventional models. Our results demonstrate CCAMT's strength in effectively leveraging the combination of physical and online activity data, and its superior performance in multi-modal depression detection.

**Comparing model convergence speed.** Figure 2a illustrates the progression of the model's validation accuracy over a 200-epoch training period. CCAMT consistently outperforms the state-of-the-art time2vec multimodal transformer (T2V) throughout this process. Notably, CCAMT learns faster than T2V, showing a steep increase in accuracy within the first 50 epochs. Our model achieves 90.4% accuracy by epoch 44, while T2V reaches its 90% accuracy at a later epoch. CCAMT is 2.8% more accurate than T2V, achieving a final accuracy of 96.4%.

Figure 2b shows the F1 score progression during training. CCAMT consistently outperforms T2V throughout the training process. Specifically, its F1 score mirrors its accuracy curve. The score begins to plateau around epoch 50 and improves by 0.063 by the final epoch. Figure 2c shows that CCAMT maintains a consistent lead in precision across all epochs, with the largest margin of 15.8% at epoch 42. The curve stabilizes after epoch 50, gaining an additional 6.4% by the end. For recall (Figure 2d), the largest gap between CCAMT and T2V is 14.6% at epoch 59. We conduct all evaluations using five-fold cross-validation. CCAMT demonstrates superior generalization and stability, evidenced by its higher average accuracy and narrower variability band across folds.

**Robustness analysis with imbalanced data.** In real-world scenarios, the number of depressed users is typically much less than the number of non-depressed users. To evaluate our model's robustness to such balance, we follow the approach used in the prior work Gui et al. (2019b). We evaluate model robustness under varying levels of data imbalance using the Multimodal Twitter dataset, which contains 1402 depressed and 1402 non-depressed users. We create imbalanced versions by randomly sampling varying percentages of depressed user data while keeping the non-depressed user data for training.

Figure 2e shows the test accuracy of CCAMT, time2vec, vanilla, and set models trained on decreasing percentage of depressed users, from 100% to 10%, while keeping the number of non-depressed users the same. Across all settings, CCAMT consistently outperforms the others, demonstrating strong robustness to data imbalance. At 90% data, it reaches 95.4% accuracy, compared to 91.4% for time2vec, 84.4% for vanilla, and 85.2% for set. As the available depressed data decreases to 50%, CCAMT still maintains a 93.0% accuracy, outperforming time2vec at 86%, vanilla at 81.4%, and set at 81.2%. At the lowest 10% level, CCAMT retains 74.2% accuracy, while time2vec, vanilla,

and set degrade significantly to 49.8%, 47.3%, and 50.2%, respectively. These results demonstrate the superior robustness of CCAMT compared to baselines, especially under severe data imbalance.

Figure 2f shows label-shuffling experiment to evaluate CCAMT's sensitivity to noise. As label noise increases, performance consistently degrades across all metrics, confirming that the model does not simply memorize labels and instead learns cross-modal relationships. Specifically, accuracy drops from 96.5% to 77% with 25% label shuffled, 76.1% at 50% shuffled, 71% at 75% shuffled, and 50.8% with 100% label shuffled. F1 also follows a similar trend, dropping from 96.4% to 80.5% with 25% label shuffled, and 66.8% with 100% label shuffled. We further provide a comprehensive ablation study in the Appendix A.2. Specifically, we conduct sensitivity analysis of alignment quality, examining how different mapping strategies influence the semantic alignment and comparing their resulting performance.

### 4.3 Edge Deployment

We deploy the models on the Jetson Nano, an IoT platform equipped with an Nvidia Maxwell GPU, 4GB memory, and 64 GB storage, to enable privacy-preserving on-device depression detection.

**Inference latency comparison.** Figure 3 compares six image and text encoder configurations to identify the lowest inference latency for CCAMT. Our chosen encoder configuration, Clip + EmoBERTa, achieves the lowest inference latency at 10.76 seconds. It achieves the lowest inference latency at 10.76 seconds, significantly faster than all other encoder configurations and up to 83.0% faster than the slowest configuration (Clip + Minilm). Text embedding generation dominates the total latency at 84.39% (9.07 seconds), followed by image embedding latency at 11.93% (1.28 seconds). The physical data encoder steps have minimal latency, with the projection and the embedding step taking only 7 ms and 80 ms. The inference time of CCAMT is only 0.22s, making CCAMT ideal for edge deployment. These results confirm that CCAMT can provide a timely prediction under resource constraints. Please refer to Appendix A.2 for additional edge deployment experiments. We evaluated various encoder configuration and showed that our selected configuration is the best for balancing latency and accuracy. We also measured different encoder size to confirm that CCAMT's encoder configuration is compact enough for edge deployment while maintaining high performance.

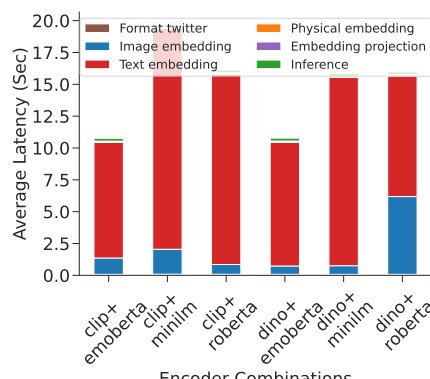

Figure 3: Inference comparison on Jetson.

## 5 Conclusions

This paper presents a two-stage framework to address training-time modality incompleteness and enable learning from datasets that combine drastically different modalities. The framework consists of Data Fusing with Label-guided Mapping (DFLM) and the Cooperative Cross-attention Multimodal Transformer (CCAMT). DFLM introduces a novel use of supervised contrastive learning to align semantically similar user data across different modalities in a shared latent space, enabling the creation of pseudo-multimodal datasets without requiring co-occurring data. CCAMT is a unique architecture designed to effectively model both intra- and inter-modality dependencies, leveraging cross-attention to capture complementary information across modalities. The proposed framework allows researchers to explore the multimodal learning benefits without collecting the co-occurring multimodal data. Our extensive evaluation results show that CCAMT consistently achieves more accurate and faster predictions than the best-published results across multiple datasets. Its deployment on real edge devices further confirms its effectiveness in resource-constrained edge environments.

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

# A APPENDIX

## A.1 IMPLEMENTATION DETAILS

We trained all models using the Adam optimizer with a base learning rate of 1e-5. The learning rate varies using a cyclical learning rate scheme, linearly varying from 1e-5 to 1e-4 every 10 epochs. We implemented our proposed method on PyTorch 1.10 and evaluated it using Nvidia RTX 2080 GPUs. For edge deployment evaluation, we used a widely used IoT platform, Nvidia Jetson Nano.

## A.2 ABLATION STUDIES

**Comparing mapping strategy.** Table 4 compares the CCAMT's performance under three mapping strategies: random, similarity-based, and contrastive-based. Our proposed contrastive mapping strategy consistently outperforms the other mapping strategies across all evaluated metrics. Despite its simplicity, the random mapping strategy yields surprisingly competitive performance (0.7% lower accuracy than contrastive mapping method), suggesting that even weak cross-modal signals can be informative.

Figure 4a to 4c show t-SNE visualizations of user embeddings under these mapping strategies. In the random mapping (Figure 4a), each online user is paired with a randomly selected physical user sharing the same label. The resulting distribution is largely unstructured, reflecting weak semantic alignment across modalities. The similarity-based mapping (Figure 4b) improves upon this by pairing each online user with the most similar physical user based on cosine similarity in embedding space. This produces better-structured semantic alignment than random mapping. The contrastive-based mapping (Figure 4c) further enhances semantic alignment by explicitly learning to bring semantically similar user pairs closer in the latent space. The resulting embeddings exhibit improved structure.

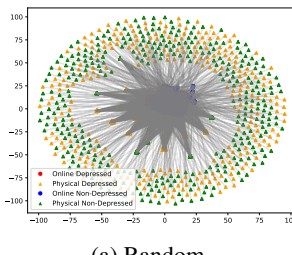 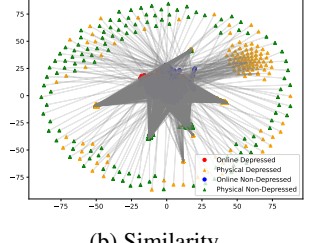 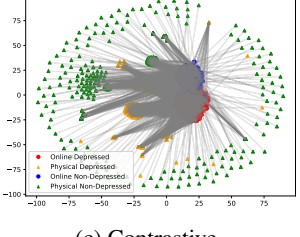

(a) Random.      (b) Similarity.      (c) Contrastive.

Figure 4: t-SNE visualizations of feature embeddings under (a) random, (b) similarity-based, and (c) contrastive-based mappings. Circles/triangles represent online/physical users; colors denote depression labels. Gray lines link the same-label synthetic modality pairs.

Table 4: Performance comparison of different mapping strategies.

|  | F1 | Recall | Precision | AUC | Accuracy |
|---|---|---|---|---|---|
| Random | 95.7 | 95.4 | 96.1 | 99.1 | 95.7 |
| Similarity | 96.0 | 95.8 | 96.3 | 99.1 | 96.0 |
| Contrastive | **96.4** | **96.0** | **96.9** | **99.2** | **96.4** |

**The impact of attention.** We evaluate the effectiveness of the cooperative cross-attention mechanism by comparing bi-directional and single-directional attention. Figure 5 illustrates the two settings within CCAMT's encoder across three modalities: image (I), text (T), and physical activity (P). In a single-directional setting, attention passes from T to P. In a bi-directional setting, it passes in both directions between T and P. We exclude attention passing between P and I, as textual data dominates the input in our multimodal datasets, making P to I passing not helpful.

Table 5 shows that bi-directional attention consistently outperforms single-directional attention across all metrics. It improves F1 score by 1.1%, recall by 1.3%, precision by 0.9%, AUC by 0.2%,

and accuracy by 0.8%. These results indicate that bi-directional attention enhances the model's overall performance, particularly in balancing precision and recall. Notably, even with single-directional attention, our method still surpasses the state-of-the-art Time2Vec transformer by 1.8%.

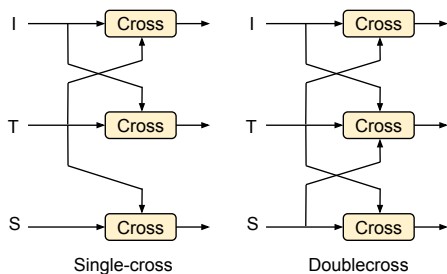

Figure 5: Cross connection.

Table 5: Single-directional vs bi-directional attention.

|                    | F1   | Recall | Precision | AUC  | Accuracy |
|--------------------|------|--------|-----------|------|----------|
| Single-directional | 95.3 | 94.7   | 96.0      | 99.0 | 95.6     |
| Bi-directional     | **96.4** | **96.0** | **96.9** | **99.2** | **96.4** |
| Improvement (%)    | 1.1  | 1.3    | 0.9       | 0.2  | 0.8      |

**The impact of window size.** Figure 6 shows the effect of varying window size, which defines the number of posts processed simultaneously, on model performance for Multimodal Twitter and Multimodal Reddit datasets. On Multimodal Twitter, increasing the window size from 32 to 512 improves most metrics. Recall shows the largest increase of 8.0% due to the model's ability to learn from more data. AUC, however, remains relatively stable with only a 2.7% change. The smaller change in AUC likely reflects its focus on class separation, making it less sensitive to window size variation than recall and accuracy. On Multimodal Reddit, the results show a similar pattern. Specifically, recall increases the most, by 17.8%. AUC changes the least, with a 10.4% difference. Overall, a larger window size increases accuracy but demands more computing resources.

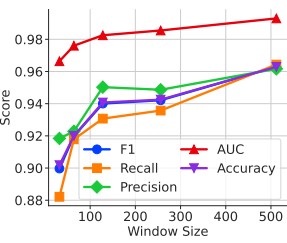
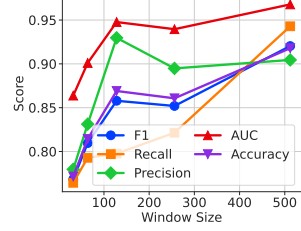

(a) Multimodal Twitter.          (b) Multimodal Reddit.

Figure 6: Comparison model performance with varying window size on (a) Multimodal Twitter and (b) Multimodal Reddit datasets.

**Encoder configuration for online activity data.** We evaluate different encoder configurations to identify the most effective setup for CCAMT's multimodal input. We fix the physical encoder and vary image-text encoder combinations. Table 6 lists the results, along with the percentage differences relative to the Clip + Emoberta baseline. Clip + EmoBERTa is the chosen encoder configuration, achieving up to 18% higher accuracy than other configurations, within 1% of the best-performing alternatives. Clip + Roberta and Dino + Roberta slightly outperform the baseline. Clip + MiniLM underperforms significantly, with a 15.3% drop in F1 score and a 16.3% in accuracy. Overall, the chosen Clip + EmoBERTa configuration improves accuracy by up to 18%, with only minor differences from the best-performing alternative.

Table 6: Performance comparison of image-text encoder configurations with a fixed physical encoder. Percentages indicate improvements over the Clip + Emoberta baseline.

| Depressed (%) | F1 | Recall | Precision | AUC | Accuracy |
|---|---|---|---|---|---|
| Clip + Emoberta | 96.4 (0.0%) | 96.0 (0.0%) | 96.9 (0.0%) | 99.2 (0.0%) | 96.5 (0.0%) |
| Clip + MiniLM | 81.2 (-15.3%) | 85.6 (-10.4%) | 77.3 (-19.6%) | 87.0 (-12.2%) | 80.2 (-16.3%) |
| Clip + Roberta | **97.3 (+0.9%)** | **96.9 (+0.9%)** | **97.7 (+0.8%)** | **99.4 (+0.1%)** | **97.3 (+0.9%)** |
| Dino + Emoberta | 96.0 (-0.5%) | 97.1 (+1.1%) | 94.9 (-2.0%) | 99.3 (-0.1%) | 95.9 (-0.5%) |
| Dino + MiniLM | 80.0 (-16.5%) | 85.9 (-10.1%) | 74.9 (-22.0%) | 85.5 (-13.7%) | 78.5 (-18.0%) |
| Dino + Roberta | **97.4 (+1.0%)** | **97.6 (+1.6%)** | **97.2 (+0.3%)** | **99.4 (+0.2%)** | **97.4 (0.9%)** |

## A.3 ADDITIONAL EDGE DEPLOYMENT EXPERIMENTS

**Inference latency and accuracy trade-off.** Clip + Emoberta achieves 96.3% F1 and 96.3% accuracy, outperforming most methods. Although its accuracy is within 1% of Clip + RoBERTa and Dino + RoBERTa, it reduces latency by up to 49.94%, making it the optimal configuration for balancing latency and accuracy.

**Size comparison.** Table 7 lists the model sizes of different encoder configurations. Our encoder configuration has a model size comparable to other configurations. Clip + EmoBERTa totals 806.41 MB, comparable to Clip + RoBERTa and only 26.67 MB larger than the smallest configuration, Clip + MiniLM (779.74 MB). Additionally, it is 2.25 MB smaller than both Dino + EmoBERTa and Dino + RoBERTa (808.66 MB), making it compact enough for edge deployment with high accuracy. These results confirm that CCAMT's encoder configuration is compact enough for edge deployment while maintaining high performance.

Table 7: Total size of various encoder configurations.

| Encoder configurations | clip + emoberta | clip + minilm | clip + roberta | dino + emoberta | dino + minilm | dino + roberta |
|---|---|---|---|---|---|---|
| Size (MB) | 806.41 | 779.74 | 806.41 | 808.66 | 781.99 | 808.66 |
| Difference (MB) | 0 | -26.67 | 0 | 2.25 | -24.42 | 2.25 |

