# OpenReview forum: "Exploring Training Time Modality Incompleteness and Learning from Diverse Modalities"
_ICLR.cc/2026/Conference — Submitted to ICLR 2026_

### Official Review · Reviewer_VXKD · 2025-10-27

**Soundness:** 2
**Presentation:** 3
**Contribution:** 3
**Rating:** 2
**Confidence:** 4

**Summary:**

The paper proposes 1) Data Fusing with Label-guided Mapping (DFLM) that aims to address missing modality during training, and 2) Cooperative Cross-attention Multimodal Transformer (CCAMT) that combines cross-attention and self-attention in learning both modality-specific and cross-modal interaction with drastically different modalities. CCAMT is extensively evaluated on 3 multimodal datasets that have both text, visual, and physical activity data. CCAMT is widely compared to several baselines to demonstrate its effectiveness. Ablation analysis also shows the robustness and efficiency in edge deployment of CCAMT.

**Strengths:**

- The paper is well-motivated by the research questions of addressing incomplete modality and multimodal learning with drastically different modalities, which are core research problems in multimodal learning with the later (multimodal learning with common modalities such as text, vision, and low-resource modalities such as physical activity data) less frequently explored in multimodal research;
- Methods and experiments are well-documented;
- The proposed method is extensively evaluated on 3 real-world tasks and compared with various baselines and SOTA methods, with reasonable margins that corroborate the effectiveness of the method;
- Ablation analysis on robustness and efficiency demonstrate the advantage and practical utility in real-world deployment;

**Weaknesses:**

The reviewer has 2 major concerns and is willing to raise scores if the authors can properly address them in the rebuttal:
- One major concern about evaluation is that given the paper proposes 2 stages, DFLM and CCAMT, while only CCAMT (or the entire pipeline) is evaluated in section 4, **the DFLM stage is not evaluated and therefore its effectiveness remains unclear**. Note there are many existing methods aiming to address the challenge of multimodal learning with missing modalities. The paper should at least include a study that ablates the effect of DFLM on the overall performance of the pipeline and compare DFLM with other techniques used to address incomplete modalities. The reviewer also recommends to name the proposed method differently from CCAMT (e.g. DFLM+CCAMT or something else), so the evaluation in section 4 is not confused with evaluation with only the CCAMT stage;
- The formulation of CCAMT resembles many existing notions of cross-modal attention such as in [1, 2, 3], so this is not a new concept and should be clearly differentiate from the existing works by highlighting the extra contribution of CCAMT on top of the existing formulation, with (empirical) evidence supporting those contribution claims;
- Writing: the writing is overall clear and easy-to-follow, but the introduction is a little too lengthy (as the method section only starts at the bottom of page 3). The introduction does not need to contain all the results, instead it should just give a high-level overview of the contributions and interesting findings.

[1] Wang, Meifang, and Zhange Liang. “Cross-modal self-attention mechanism for controlling robot volleyball motion.” Frontiers in neurorobotics vol. 17 1288463. 10 Nov. 2023, doi:10.3389/fnbot.2023.1288463

[2] Tsai, Jia-Hua, and Wei-Ta Chu. “Multimodal Fusion with Cross-Modal Attention for Action Recognition in Still Images.” Proceedings of the 4th ACM International Conference on Multimedia in Asia (MMAsia ’22), Association for Computing Machinery, 2022, Article 31, 5 pp.

[3] X. Wei, T. Zhang, Y. Li, Y. Zhang and F. Wu, "Multi-Modality Cross Attention Network for Image and Sentence Matching," 2020 IEEE/CVF Conference on Computer Vision and Pattern Recognition (CVPR), Seattle, WA, USA, 2020, pp. 10938-10947, doi: 10.1109/CVPR42600.2020.01095.

**Questions:**

- Weakness 1: what is the effect of DFLM? How is its effectiveness compared to other existing approaches in addressing multimodal learning with missing modalities?
- Weakness 2: how is CCAMT different from existing formulation of cross-modal attention?

---

> ### Author Response · Authors · 2025-11-21
>
> Dear Reviewer VXKD:
>
> We thank the reviewer for raising these important points and respond to them below.
>
> **W1: Comparison to other techniques for missing modalities.**
> Existing methods for missing modalities typically assume co-occurring multimodal data and focus on (1) hallucinating missing modalities at test time or (2) training models that remain functional with incomplete training samples. Examples include hallucination/distillation methods (Hoffman et al., García et al.), contrastive alignment for modal-incomplete robustness (Lin et al.), and multi-task or Bayesian approaches that handle partially missing modalities during training (Fortin et al., Ma et al.).
>
> In contrast, DFLM addresses a fundamentally different scenario: training-time modality incompleteness with no co-occurring multimodal data at all. Instead of relying on paired samples or hallucinating missing features, DFLM constructs pseudo-multimodal samples by aligning heterogeneous single-modality datasets through label-guided contrastive learning. This enables us to learn from diverse modalities collected from disjoint user groups, a setting where existing missing-modality methods cannot be applied because they require at least some paired multimodal examples to learn cross-modal correspondences.
>
> The effect of DFLM is that it creates a usable multimodal training set from originally disconnected datasets, making it possible for CCAMT to perform multimodal fusion and achieve performance gains.
>
> [Hoffman et al.] Learning with side information through modality hallucination
>
> [García et al.] Modality distillation with multiple stream networks for action recognition
>
> [Lin et al.] Missmodal: Increasing robustness to missing modality in multimodal sentiment analysis
>
> [Fortin et al.] Multimodal Sentiment Analysis: A Multitask Learning Approach
>
> [Ma et al.] Smil: Multimodal learning with severely missing modality
>
>
> **W2: CCAMT’s difference from existing cross-modal attention.**
>
> CCAMT is different from existing cross-modal attention in the following to aspects.
> First, CCAMT incorporates a third branch for the unified physical activity modality, which aggregates multiple heterogeneous sensor modalities into a single sequence through our customized, lightweight transformer.
> Second, CCAMT uses selective bidirectional attention: cross-attention flows between Text to Physical and Image to Text, but not Image to Physical. We intentionally exclude Physical to Image attention empirically because text dominates the input in our data modalities. Please refer to Appendix A.2 Impact of attention for details.

---

> > ### Comment · Reviewer_VXKD · 2025-11-22
> >
> > Thanks the authors for their clarifications. The reviewer now agrees that the label-guided alignment with contrastive learning is novel but still has the following concerns:
> >
> > - Effectiveness of DFLM on the final performance i.e. Appendix Table 4 and Figure 4 should be highlighted in the main text;
> > - The "selectiveness" of CCAMT is only lightly touched in line 244 in the main text (and later illustrated in Appendix A.2 as a part of the ablation) and is not highlighted in the formulation presented in section 3.2, which follows the existing formulation of cross-modal attention in a lot of prior work. Moreover, regarding the authors' response about how CCAMT differentiates from existing cross-modal attention: 1. the notion of cross-modal attention exists as a general attention mechanism that applies to different modalities, which include physical data, so this should not be the novelty of CCAMT, unless the authors clarify how physical data is not compatible with the existing cross-modal attention formulation and only applicable to CCAMT; 2. the selection criteria seems to be decided by domain knowledge / empirically ("We intentionally exclude Physical to Image attention empirically because text dominates the input in our data modalities") and does not seem to be a novel contribution of CCAMT.

---

> ### Author Response · Authors · 2025-11-26
>
> Dear Reviewer VXKD:
>
> Thank you for your response.
> - We will highlight Appendix Table 4 and Figure 4 in the revision.
> - We will clarify the "selectiveness" of CCAMT in the revision.

---

### Official Review · Reviewer_DrjM · 2025-10-28

**Soundness:** 2
**Presentation:** 3
**Contribution:** 2
**Rating:** 2
**Confidence:** 4

**Summary:**

This paper addresses the challenge of modality incompleteness during training in multimodal learning through a two-stage framework. In the first stage, **D**ata **F**using with **L**abel **G**uided **M**apping (**DFLM**), the model constructs pseudo-multimodal datasets by pairing users from separate single-modal datasets (online activity and physical activity) using supervised contrastive learning based on shared depression labels. In the second stage, **C**ooperative **C**ross **A**ttention **M**ultimodal **T**ransformer (**CCAMT**), it integrates cross-attention for inter-modal interaction and self-attention for intra-modal representation learning. Experimental results demonstrate consistent improvements over baselines and show the feasibility of the proposed approach for edge deployment.

**Strengths:**

- The paper provides comprehensive experimental results and analysis supporting its main contributions
- The proposed approach is evaluated on a variety of tasks, including general applications (e.g., classification) as well as deployment on edge devices, demonstrating its broad applicability.

**Weaknesses:**

1. The main contribution of this paper appears somewhat incremental. Specifically, there already exist several prior works that incorporate both cross-attention and self-attention mechanisms during training to effectively leverage intra- and inter-modal information [1][2][3].
2. The comparisons in Table 1 and Table 2 seem potentially unfair due to differences in the number of modalities. Moreover, the proposed model shows slightly degraded performance under equivalent settings. For instance, on the Twitter dataset, Time2VecTransformer very slightly outperforms CCAMT w/o physical (T+I modalities) in terms of F1, Precision, and Accuracy.
3. In my understanding, the paper lacks clear justification and consistency in its analysis settings. For example:
    - The robustness analysis in Section 4.2 (particularly Fig.2e) focuses on class imbalance between depressed and non-depressed categories. However, in recent works, robustness typically refers to Out-of-Distribution (OOD) generalization or adversarial settings. This may lead to confusion regarding the interpretation of “robustness.”
   - Fig.2f presents the degradation of performance under label shuffling but does not clearly demonstrate whether CCAMT is robust to label noise. The message conveyed is thus unclear.
   - The edge-device analysis in Section 4.3 appears to hinder the paper’s main contribution, as it lacks comparative evaluations against other architectures to substantiate whether CCAMT achieves superior inference efficiency.

[1] Wei, Xi, et al. "Multi-modality cross attention network for image and sentence matching." CVPR 2020\
[2] Quan, Weize, et al. "TCAN: Text-oriented cross attention network for multimodal sentiment analysis." Preprint 2025\
[3] Zhang, Zhicheng, et al. "Moda: Modular duplex attention for multimodal perception, cognition, and emotion understanding." ICML 2025

**Minor Concern** (not affect the rating)
- The combination of Online Activity datasets (Twitter/Reddit) and Physical Activity datasets (StudentLife) is unconventional and not directly comparable to such general multimodal benchmarks (e.g., CREMA-D, Kinetics-Sounds, CMU-MOSEI). In my point of view, this makes it slightly difficult to assess how CCAMT would perform on widely-used evaluation protocols or to compare against the broader multimodal learning.
- Typos
  - line 198:  online embedding $\ z_i^{\text{online}}\$} $\rightarrow$ {$\ z_i^{\text{online}}\$}
  - Eq.(2), Eq.(3): The font size appears inconsistent and should be standardized.

**Questions:**

- **Clarification on Weaknesses 1**: How does the proposed method differ fundamentally from existing attention-based multimodal frameworks?
- **Clarification on Weaknesses 2**: How was fairness ensured in the comparisons, particularly regarding the number of modalities used? Does this imply that the improvements are primarily due to access to additional modality data rather than the effectiveness of the proposed approach itself?
- **Clarification on Weaknesses 3**: How do the authors define “robustness” in this context, and how does it relate to standard OOD or adversarial evaluations?
- Have the authors considered validating CCAMT on other multimodal learning tasks beyond classification, such as retrieval or visual question answering (VQA)?
- Additional experiments could further substantiate the robustness claims. For instance, incorporating adversarial attack evaluations would strengthen the paper that CCAMT is a robust model [1][2][3]

[1] Yin, Ziyi, et al. "Vlattack: Multimodal adversarial attacks on vision-language tasks via pre-trained models." NeurIPS 2023\
[2] Dou, Zhihao, et al. "Adversarial attacks to multi-modal models." ACM Workshop LAMPS 2024\
[3] Cui, Xuanming, et al. "On the robustness of large multimodal models against image adversarial attacks." CVPR 2024\

=======================================================

**Note**: I acknowledge that I may have partially misunderstood certain aspects of the paper. Therefore, I am willing to raise my rating score if these questions and concerns are adequately addressed.

---

> ### Author Response · Authors · 2025-11-21
>
> Dear Reviewer DrjM,
>
> We thank the reviewer for the valuable comments and suggestions. Our responses are provided below.
>
> **W1: Incremental contribution.**
> Please refer to the response to **Reviewer cmNd W1.**
>
> **W2: Unfair comparison.**
> The core contribution of our work is the two-stage cooperative multimodal framework (DFLM + CCAMT), which enables multimodal learning without co-occurring modalities. This is a setting fundamentally different from the one assumed by standard dual-modality architectures such as Time2VecTransformer.
>
>
> **W3.1: Definition of robustness.**
> We will clarify that the robustness analysis is about data imbalance in the revision. This type of analysis is commonly used in depression detection literatures [1].
>
> [1] Gui, Tao, et al. "Cooperative multimodal approach to depression detection in twitter." Proceedings of the AAAI conference on artificial intelligence. Vol. 33. No. 01. 2019.
>
> **W3.2: Label noise.**
> The purpose of Fig. 2f is to confirm that our model is not overfit to labels instead of learning the intended cross-modal relationships. Full label shuffling serves as a stress test showing that performance degrades sharply, confirming that the model does not simply memorize labels and does depend on meaningful label-guided alignment. We will clarify this in the revision.
>
> **W3.3: Edge-analysis hinders main contribution.**
> The edge-device analysis verifies that our cooperative multimodal framework remains lightweight enough for deployment on a real and  resource-constrained edge device.
>
> **Minor: Unconventional dataset combination.**
> Our setting fundamentally differs from the standard co-occurring multimodal benchmarks used in CREMA-D, Kinetics-Sounds, or CMU-MOSEI. These datasets assume strong temporal or instance-level correspondence between modalities (e.g., audio–video–text for the same utterance). In contrast, our work addresses the cooperative scenario where modalities originate from disjoint user groups and never co-occur, which cannot be evaluated on these benchmarks.
>
> The combination of Twitter/Reddit and StudentLife is chosen precisely because this domain provides naturally non-co-occurring online and physical modalities, making it one of the few available real-world settings where our formulation is meaningful. We will clarify that the goal of this work is not to compete on standard multimodal benchmarks, but to introduce and evaluate a framework for multimodal learning when co-occurrence does not exist.

---

> ### Comment · Reviewer_DrjM · 2025-11-25
> **Response to the Authors' Rebuttal**
>
> Thank you for the clarifications. While many of my concerns have been addressed, I still have one major remaining issue.
>
> As **Reviewer VXKD** mentioned, the key weakness regarding the absence of experiments with three modalities has not been fully resolved. The paper demonstrates that adding more modalities leads to clear improvements, yet there is no comparison on 3-modality results for the Twitter and Reddit datasets. In my view, additional results evaluating performance with three modalities are necessary. Without such evidence, it becomes difficult to assess the contribution of the proposed two-stage cooperative training procedure, especially given that it requires additional modalities compared to other approaches. This substantially and obviously weakens the main novelty of this paper and practical significance of the method.
>
> As a minor point, I believe it would be beneficial to report the similarity values in Appendix A.2 (Ablation Studies) to better contextualize the differences among the random, similarity-based, and contrastive-based mappings.

---

> > ### Author Response · Authors · 2025-12-04
> >
> > Dear Reviewer DrjM,
> >
> > Thank you for your response.
> > We added the comparison on three modality results.
> > Following table shows the comparison. CCAMT consistently outperforms MulT [1] across both datasets.
> > We will add the new results to the revision.
> > | Dataset | Method | F1   | Recall | Precision | Accuracy |
> > |---------|--------|------|--------|-----------|----------|
> > | Twitter | MulT   | 95.6 | 95.6   | 95.5      | 95.6     |
> > |         | CCAMT  | **96.4** | **96.0**   | **96.9**      | **96.5**     |
> > | Reddit  | MulT   | 92.2 | **94.6**   | 89.8      | 92.0     |
> > |         | CCAMT  | **93.6** | 93.0   | **94.2**      | **93.5**     |
> >
> >
> > MulT is a multimodal transformer that aligns and fuses information across modalities through cross-modal attention.
> > To include MulT as a three-modality baseline, we implemented it within our two-stage pipeline.
> > The original MulT targets synchronized audio–video–text streams and cannot be directly integrated into our framework because:
> > 1) our modalities come from heterogeneous encoders and therefore lack the consistent low-level temporal structure assumed in MulT, and
> > 2) our pipeline operates on sequences are resampled into a unified temporal window with a fixed masking format that MulT does not natively support.
> >
> > To address this, we preserved its core components, including temporal convolution alignment and directional cross-modal attention. We then convert each modality’s embeddings into MulT's expected input structure and applied our unified fixed-window masks to make its alignment layers operate correctly within our pipeline. MulT and applying our unified fixed-window masks, ensuring that its alignment layers operate correctly within our pipeline.
> >
> > [1] Tsai, Yao-Hung Hubert, et al. "Multimodal transformer for unaligned multimodal language sequences."

---

### Official Review · Reviewer_PXWf · 2025-10-30

**Soundness:** 3
**Presentation:** 3
**Contribution:** 3
**Rating:** 6
**Confidence:** 3

**Summary:**

In this work, the authors proposed a novel pipeline that allows incorporation of drastically different data from another modality without co-occurrence to improve multimodal classification performance. Specifically, the work explored incorporating physical data into online data for depression detection. The proposed method first uses DFLM to pseudo-align each online user with some physical data, then devised a new multimodal fusion architecture called CCAMT that has both cross-modal attention and self attention within each modality. The proposed method was evaluated on two depression datasets with online content, and used one physical activity-depression dataset as auxiliary physical modality, and the paper showed that the proposed method outperforms all baselines, and that incorporating physical data improves performance. Additional analysis was performed to demonstrate the proposed method's robustness to fewer training data and incomplete modalities.

**Strengths:**

1. The paper explored a very interesting direction of incorporating un-aligned data from a completely different modality (but about the same task) to improve model performance on the task. The idea is innovative, and the experiments demonstrated its validity.

2. The proposed fusion module, CCAMT, although simple, achieved strong performance against baseline fusion models.

3. There is extensive analysis on the proposed method's robustness under partial training data or modality imbalance, and it is also nice to have the encoder-choice latency study to help future researchers select their encoders.

**Weaknesses:**

1. The proposed method is only designed for and evaluated on one particular domain/task: depression detection. Thus, it is completely unclear whether this idea of incorporating a very different modality on the same task without co-occurrence can be applied to any other multimodal domains or tasks at all, and the application generalizability of this method may be rather limited.

2. It is not very clear to me why the contrastive approach for DFLM works. It seems like the contrastive objective is trained over randomly paired data within each label group over the same set of data as the one we try to pair with similarity later, so if the projection modules are trained to convergence with the InfoNCE objective, wouldn't the projection layers just memorize the random pairings, and the similarity-based mapping would just be identical to the random pairings?

**Questions:**

1. In line 174, all 3 X have different sequence lengths, but after passing them through pre-trained encoders, all of a sudden all 3 modalities have sequence length T. How does this happen? I don't think the arbitrarily selected pre-trained encoders (like the ones in the experiments) can guarantee that all 3 modalities have the same encoded sequence length. Do you perform the 1D interpolations here instead of later (as in lines 192-193)?

2. Following the question above, if the sequence lengths are already matched to T in line 174 for the three E, then the z in line 190-191 will always have matched sequence length. Is the 1D linear interpolations (on line 192-193) still necessary?

---

> ### Author Response · Authors · 2025-11-21
>
> Dear Reviewer PXWf,
>
> We appreciate your constructive feedback, which we respond to in detail below.
>
> **W1: Limited generalizability.**
> Most existing multimodal sentiment datasets (e.g., MOSI, MOSEI, CMU-MOSEAS) provide co-occurring modalities such as aligned video, audio, and text. This setup differs from the non-co-occurring, cross-user scenario we study, where modalities come from separate sources and share only label-level supervision. As a result, these datasets do not naturally support evaluating our cooperative alignment setting, and we will clarify this distinction in the revision.
>
> Although our experiments focus on depression detection, the underlying methodology is not limited to this domain. The core contribution is a general two-stage framework—DFLM for label-guided alignment of non-co-occurring modalities and CCAMT for their cooperative fusion. Both components are modality-agnostic and operate on generic latent sequences without relying on domain-specific assumptions. Depression detection is used as a concrete use case, and we will clarify this in the revision and note that applying the framework to additional domains is an important direction for future work.
>
> **W2: Effectiveness of DFLM.**
> Appendix Figure 4 and Table 4 provide additional sensitivity analysis on the alignment quality. Figure 4 visualizes how different mapping strategy changes the semantic alignment and Table 4 compares the performance of these mapping strategies.
>
>
> **Q1 and Q2:**
> The text and image modalities are preprocessed so that their encoders always produce a fixed target sequence length. The physical modality does not naturally match this length, so we apply a single 1D interpolation step to resize its sequence to match this length. We will clarify this in the revision.

---

> > ### Comment · Reviewer_PXWf · 2025-11-25
> >
> > Thank you for your clarifications! My scores remain positive.

---

> > > ### Author Response · Authors · 2025-11-26
> > >
> > > Dear Reviewer PXWf,
> > >
> > > Thank you for your response. I appreciate your time.

---

### Official Review · Reviewer_cmNd · 2025-11-01

**Soundness:** 2
**Presentation:** 2
**Contribution:** 2
**Rating:** 6
**Confidence:** 2

**Summary:**

This paper proposes a two-stage framework to address the highly practical challenge of training-time modality incompleteness, where real-world modalities (like social media and wearable data) are often collected from different, non-overlapping user cohorts. To address this, the authors propose to perform Data Fusing with Label-guided Mapping (DFLM), which uses shared labels (e.g., "depressed") and Supervised Contrastive Learning to map disparate modal embeddings into a shared latent space. To effective learn from the generated pseudo multimodal datasets, a cooperative cross-attention transformer is presented. Comprehensive evaluations across diverse datasets and deployment settings, demonstrating the effectiveness and robustness of our framework.

**Strengths:**

- The paper is easy to follow and the proposed solution is technically sound;
- Implementation details, hyperparameters, and encoder combinations are clearly reported;
- The combination of label-guided contrastive alignment and cross-attention fusion  is conceptually coherent.

**Weaknesses:**

Weakness & Questions:
- Cross-attention fusion. It is common to use Cross-attention for multimodal fusion, and CCAMT’s architecture is a bit similar to standard cross-attention Transformers with bidirectional connections (Multimodal Transformer). Are there any specific designs in CCAMT?
- There’s no analysis of failure cases. For example, when modalities are weakly correlated or when label noise disrupts alignment.
- For better verify the effectiveness of the proposed approach, please conclude more baselines for comparison, e.g., CycleGAN-style modality transfer, CMMD (contrastive-based);
- Running efficiency comparison. Please also include more quantitative evaluation of inference & running efficiency.

**Questions:**

Please see weakness.

---

> ### Author Response · Authors · 2025-11-21
>
> Dear Reviewer cmNd:
>
> We appreciate the reviewer’s insightful comments and the suggestion on important related works.
> We provide detailed clarifications and responses below.
>
>
> **W1: Novelty of Cross-attention fusion.**
> The novelty of our work lies in the two-stage design that enables multimodal learning when modalities come from different users and datasets. Existing multimodal Transformers assume fully paired data and therefore cannot be applied directly in our setting.Specifically, our method is novel in two ways:
>
> 1. A new cooperative multimodal learning setting enabled by DFLM + CCAMT.
> DFLM generates label-guided pseudo-multimodal datasets between online and physical data. CCAMT learns from these datasets. The two stages allow us to explore the training-time modality incompleteness problem.
>
> 2. CCAMT is a cross-attention–based fusion module designed to integrate DFLM-aligned online and physical modalities through selective bidirectional attention tailored to our cooperative multimodal setting. Specifically, CCAMT integrates a unified physical activity branch created by aggregating multiple sensor modalities with a lightweight transformer. It then applies selective bidirectional attention by allowing interactions between Text and Physical and between Image and Text, while discarding direct interaction between Image and Physical; this configuration was chosen based on empirical observations that textual data dominates the input in our datasets and is further discussed in Appendix A.2.
>
> **W2: Lack of analysis of failure cases.**
> Figure 2f reports performance under label shuffling, which a form of matching noise that disrupts the label-guided alignment. The resulting degradation confirms that the model depends on meaningful cross-modal relationships rather than overfitting to labels. Additionally, Appendix A.2 provides a sensitivity analysis of alignment quality. Figure 4 illustrates how different mapping strategies affect semantic alignment and Table 4 shows how performance degrades under these mapping strategies.
>
> **W3: Compare with more baselines.**
> CycleGAN-style modality transfer is not directly applicable to our setting. CycleGAN-style modality transfer assumes pixel- or feature-level domain translation between two domains that represent the same underlying content (e.g., image-to-image translation).
> In our case, the online and physical modalities originate from different users and capture fundamentally different types of signals, making direct translation between them infeasible. Therefore, CycleGAN-based translation methods are not directly suited for our cross-user, cross-modality data construction task. CMMD and our method both leverage contrastive learning, but they address fundamentally different multimodal problems. CMMD  operates in a paired, temporally aligned video–audio setting where each training example contains both modalities synchronized in time. Although the raw signals are aligned, the latent representations produced by video and audio encoders are not guaranteed to be semantically aligned. CMMD therefore introduces a contrastive diffusion loss to encourage semantic alignment of already co-occurring modalities inside a generative diffusion model.
>
> In contrast, our framework addresses a completely different challenge: multimodal learning when modalities do not co-occur and come from disjoint user groups. DFLM uses a contrastive objective to to align semantically similar user data across heterogeneous single-modal datasets and construct a pseudo-multimodal dataset. We will clarify this in the revision.
>
> **W4: Running efficiency comparison.**
> Please refer to the Appendix A.3. for additional edge deployment experiments. We evaluated various encoder configuration and showed that our selected configuration is the best for balancing latency and accuracy. We also measure different encoder size to confirm that CCAMT’s encoder configuration is compact enough for edge deployment while maintaining high performance.

---

### Author Response · Authors · 2025-12-04

We sincerely appreciate the time and effort the reviewers have invested in evaluating our submission. We have carefully addressed all comments and incorporated the corresponding revisions into the manuscript (highlighted in blue).

---

### Meta-Review · Area_Chair_ny2v · 2026-01-07

**Summary:**

reviewers gave 2,2,6,6, with the main weaknesses being:

Incremental contribution: there already exist several prior works that incorporate both cross-attention and self-attention mechanisms during training to effectively leverage intra- and inter-modal information

Experimental concerns: comparisons in Table 1 and Table 2 potentially unfair due to differences in the number of modalities. Moreover, the proposed model shows slightly degraded performance under equivalent settings. Paper also lacks clear justification and consistency in its analysis settings, whether class imbalance, label shuffling, OOD, label noise, or edge-device analysis. Parts where motivation for experimental setting is not clearly written.

Methodology and ablations: the paper proposes 2 stages, DFLM and CCAMT, while only CCAMT (or the entire pipeline) is evaluated in section 4, the DFLM stage is not evaluated and therefore its effectiveness remains unclear. lack of baselines to multimodal learning with missing modalities.

**Reviewer Concerns:**

Incremental contribution

--> somewhat addressed by motivating the necessity for 2 stage approach, but lack of ablations and rigorous eval of the 2 stages remain a problem if this is the motivation

Experimental concerns and lack of baselines

--> gave some reasons why other baselines are not relevant but did not add any more experiments

Analysis settings, whether class imbalance, label shuffling, OOD, label noise, or edge-device analysis. Parts where motivation for experimental setting is not clearly written.

--> somewhat addressed, add some analysis in various settings, but overall clarity concerns remain.

Methodology and ablations:

--> somewhat addressed

**Reviewer Scores:**

Both reviewers who gave 2 might have increased to 4 but no higher in my estimate.

---

### Decision · Program_Chairs · 2026-01-26

Reject